# Polypharmacy and the Change of Self-Rated Health in Community-Dwelling Older Adults

**DOI:** 10.3390/ijerph20054159

**Published:** 2023-02-25

**Authors:** Muhammad Helmi Barghouth, Elke Schaeffner, Natalie Ebert, Tim Bothe, Alice Schneider, Nina Mielke

**Affiliations:** 1Charité–Universitätsmedizin Berlin, corporate member of Freie Universität Berlin and Humboldt Universität zu Berlin, Institute of Public Health, Charitéplatz 1, 10117 Berlin, Germany; 2Charité–Universitätsmedizin Berlin, corporate member of Freie Universität Berlin and Humboldt Univer-sität zu Berlin, Institute of Biometry and Clinical Epidemiology, Charitéplatz 1, 10117 Berlin, Germany

**Keywords:** polypharmacy, self-rated health change, older adults, epidemiology

## Abstract

Polypharmacy is associated with poorer self-rated health (SRH). However, whether polypharmacy has an impact on the SRH progression is unknown. This study investigates the association of polypharmacy with SRH change in 1428 participants of the Berlin Initiative Study aged 70 years and older over four years. Polypharmacy was defined as the intake of ≥5 medications. Descriptive statistics of SRH-change categories stratified by polypharmacy status were reported. The association of polypharmacy with being in SRH change categories was assessed using multinomial regression analysis. At baseline, mean age was 79.1 (6.1) years, 54.0% were females, and prevalence of polypharmacy was 47.1%. Participants with polypharmacy were older and had more comorbidities compared to those without polypharmacy. Over four years, five SRH-change categories were identified. After covariate adjustment, individuals with polypharmacy had higher odds of being in the stable moderate category (OR 3.55; 95% CI [2.43–5.20]), stable low category (OR 3.32; 95% CI [1.65–6.70]), decline category (OR 1.87; 95% CI [1.34–2.62]), and improvement category (OR 2.01; [1.33–3.05]) compared to being in the stable high category independent of the number of comorbidities. Reducing polypharmacy could be an impactful strategy to foster favorable SRH progression in old age.

## 1. Introduction

In 2021, the proportion of individuals in Germany older than 65 years reached 22.1% of the entire population and is projected to further increase in the future [1]. A similar trend has also been predicted for the rest of Europe and, for example, the USA [2]. Such rise is accompanied by an increased burden of multimorbidity [3].

In treating multiple comorbidities, the adherence to evidence-based guidelines of disease management usually involves the concurrent prescription of multiple medications [4,5]. The resulting polypharmacy and potential drug–drug interactions are associated with an increased risk of adverse events [6]. Such adverse events include falls, hospitalization, and adverse drug interactions, which negatively affect quality of life and lead to functional impairment [5,7,8,9].

Despite the concurrent intake of multiple medications being a widespread phenomenon, there is still no agreement on a definition of polypharmacy [10]. The most commonly used definition of polypharmacy utilizes a numerical cut-off value of five medications [10]. 

Self-rated health (SRH) is one of the patient-reported outcomes (PRO) which represents a complex subjective measure of the individual’s overall health status and encompasses biological and psychosocial aspects of health [11,12]. SRH is usually measured using a single item through asking participants to rate their health from very poor to very good [13]. Although it is a subjective assessment, it is considered a reliable and valid health indicator of healthcare service utilization and mortality in older adults, and it also reflects the patient’s voice [14,15]. The simplicity and ease of implementation provided by the single-item SRH warrant its use in practice for assessment of SRH in large samples of older adults when time efficiency is of importance [16,17].

Several studies reported an association between polypharmacy and poor SRH, however, only in a cross-sectional design [18,19]. SRH progression reflects changes in physical and mental health of older adults over time and possibly before the diagnosis of disease [20]. However, the change in subjective assessment of health with advancing age in individuals with polypharmacy has not received sufficient attention [21].

Thus, the aim of this study was to (1) identify categories of SRH-change over the period of four years and describe how they differ between older individuals with and without polypharmacy and (2) assess the association between polypharmacy and different SRH-change categories. 

## 2. Methods

### 2.1. Study Population

This study utilized data from the Berlin Initiative Study (BIS). The BIS is a population-based prospective cohort of 2069 community-dwelling older individuals. Participant recruitment took place between November 2009 and July 2011 with four subsequent biennial follow-up visits. The concept and design of the BIS are described elsewhere [22]. To partake in the study, individuals had to be at least 70 years old and a member of the *Allgemeine Ortskrankenkasse (AOK)—Nordost* statutory health insurance fund. Dialysis patients and kidney transplant patients at baseline were excluded. 

For the current analysis, data from the BIS baseline visit (2009–2011) and the BIS second follow-up visit (2014–2015) (hereinafter referred to as the SRH follow-up visit) were used. The inclusion criteria were answering the SRH question at both visits and provision of information about the polypharmacy status at the BIS baseline visit. Of the 2069 participants taking part in the BIS baseline visit, eight participants were excluded due to missing SRH, and three participants were excluded due to missing information about their polypharmacy status. Between both study visits, 361 participants died, and 268 were lost to follow-up. At the SRH follow-up visit, one participant was excluded due to missing SRH, yielding a final sample of 1428 participants (Appendix A). 

### 2.2. Measures

#### 2.2.1. Polypharmacy Status

Medication was assessed by asking participants to bring their medications and medication lists to the study visit in addition to self-reporting their medication intake. Prescription and over-the-counter (OTC) medications were recorded and entered into the standardized questionnaire linked to a drug database in which drug information including the prescription requirement as well as the anatomic therapeutic chemical (ATC) code were automatically assigned. Polypharmacy status was then recorded as a binary variable, where participants regularly taking five or more medications concurrently were regarded as individuals with polypharmacy, whereas those regularly taking less than five medications or none were regarded as individuals without polypharmacy [10]. 

#### 2.2.2. Self-Rated Health (SRH)

Participants were asked to rate their health by answering the question “In general, how do you rate your health condition?” with five possible responses: very good, good, moderate, poor, and very poor. This measure has been widely used in previous studies [23,24]. As only few participants rated their health as very good or very poor (Appendix A), these two categories were combined with good and poor, respectively, yielding an aggregated three-category variable: good, moderate, and poor. 

#### 2.2.3. Determination of SRH-Change Categories

SRH progression was assessed by observing the change in SRH between both visits, through which five SRH-change categories were identified. Participants were assigned to the (1) *stable high* category when they reported their SRH as good during both visits; (2) *stable moderate* category when they reported their SRH as moderate during both visits; (3) *stable low* category when they reported their health as poor during both visits; (4) *decline* category when their SRH-changed from moderate to poor or from good to moderate or poor between visits; and lastly, (5) *improvement* category when their SRH-changed from moderate to good or from poor to moderate or good between visits.

#### 2.2.4. Covariable Assessment

During the baseline assessment of the participants, data on demographics, lifestyle factors, comorbidities, and medications were gathered by way of a standardized questionnaire in addition to anthropometric and clinical assessments. The collected data were augmented by the AOK claims data in which all comorbidities were listed according to the Tenth Revision of the International Classification of Diseases (ICD-10), which corroborated self-reported data. 

Data about the following sociodemographic variables were collected: age, gender, monthly income as a categorical variable (<1000, 1000–1999, ≥2000 (EUR)); general and vocational education as low, intermediate, or high according to the Comparative Analysis of Social Mobility in Industrial Nations (CASMIN) scale [25]; and lastly, partner status as a dichotomous variable. 

Lifestyle factors were recorded as categorical variables: frequency of alcohol consumption (less than once a month, twice or less per week, and regular consumption); physical activity (less than once a week, 1–5 times a week, or more than 5 times a week); and body mass index (BMI) (<25, 25– < 30, ≥30 kg/m^2^).

To determine the participants’ burden of multimorbidity, the Charlson Comorbidity Index [26] was used based on the information derived from the AOK claims data. The CCI comprises the following comorbidities: myocardial infarction, congestive heart failure, peripheral vascular disease, cerebrovascular disease, dementia, chronic pulmonary disease, connective tissue disease, peptic ulcer disease, liver disease, diabetes, hemiplegia, renal disease, diabetes with end organ damage, tumors, leukemia, lymphoma, and AIDS. The aforementioned conditions were then assigned clinical weights according to their adjusted one-year mortality risk while controlling for the severity of the conditions. The total CCI score is the sum of the individual weights of constituent comorbidities, with higher scores corresponding to a greater disease burden and mortality risk. For the purpose of the current study, the CCI was operationalized as a categorical variable based on the number of comorbidities including the following categories: 0, 1–2, 3–4, and ≥5 comorbidities.

### 2.3. Statistical Analyses

The characteristics of the study population stratified by (1) polypharmacy status as well as (2) polypharmacy status and SRH-change category were described by using means and standard deviations or medians and interquartile ranges for continuous variables. For categorical variables, absolute and relative frequencies were reported. To address the first research question, descriptive statistics were used to summarize the baseline characteristics of the SRH-change categories stratified by polypharmacy status. To address the second research question, multinomial regression analysis was used to compute crude and adjusted odds ratios (OR) and 95% confidence intervals (95% CI) for the association between polypharmacy and the SRH-change categories. The multinomial regression model was adjusted for a knowledge-based set of variables consisting of gender, income, CASMIN, partner status, physical activity, frequency of alcohol consumption, BMI, and CCI categories [27]. In order not to overlook age as a potential confounder, we performed an additional multinomial regression analysis containing the aforementioned variables in addition to age as the adjustment set. 

As a sensitivity analysis, the main analysis was stratified by gender to investigate its potential effect on the association between polypharmacy and SRH-change categories. All analyses were conducted using the R statistical software (Version 4.1.1; R Foundation for Statistical Computing, Vienna, Austria). 

## 3. Results

### 3.1. Main Characteristics of the Study Population

Table 1 shows the main baseline characteristics of the study population (N = 1428) stratified by polypharmacy status. The mean (SD) age was 79.1 (6.1) years; 54.0% were females; and 54.1% rated their health as good, 37.6% as moderate, and 8.3% as poor. Regarding lifestyle factors, 42.5% consumed alcohol less than once per month, 30.3% were physically active more than five times per week, and 26.9% had a BMI ≥30 kg/m^2^. One-third (34.7%) of the participants were living with five or more comorbidities. The prevalence of polypharmacy was 47.1% (N = 672). In participants without polypharmacy, 68.7, 27.1, and 4.2% rated their health as good, moderate, and poor, respectively, whereas in participants with polypharmacy, 37.8, 49.4, and 12.8% rated their health as good, moderate, and poor, respectively. Participants without versus with polypharmacy were more frequently regular alcohol consumers (22.0 vs. 19.3%), more likely to be physically active more than five times a week (36.4 vs. 23.5%), and had less often a BMI ≥30 kg/m^2^ (21.3 vs. 33.2%). Regarding CCI, 18.0% of the participants without polypharmacy were living with five or more comorbidities compared to 53.6% of the participants with polypharmacy. When comparing individuals who were excluded to study participants (Appendix A), excluded individuals were found to be older (mean age (SD) 83.3 (7.1) vs. 79.1 (6.1)) and less often females (49.5 vs. 54.0%). At baseline, a lower proportion of excluded individuals reported their SRH as good (45.6 vs. 54.1%), and a higher proportion reported their SRH as poor (13.9 vs. 8.3%). Furthermore, excluded individuals had a higher prevalence of polypharmacy (56.8 vs. 47.1%), and a higher proportion of them were living with at least five comorbidities (48.5% vs. 34.7%).

### 3.2. SRH Transition by Polypharmacy Status

The transition in SRH reporting between both study visits stratified by polypharmacy is displayed in Figure 1. In individuals without polypharmacy, the proportion of individuals reporting good SRH decreased from 68.7 at baseline to 56.0% at follow-up, whereas the proportion of individuals reporting moderate and poor SRH increased from 27.1 and 4.2% at baseline to 32.3 and 11.8% at follow-up, respectively. On the other hand, in individuals with polypharmacy, the proportion of individuals reporting good and moderate SRH decreased from 37.8 and 49.4% at baseline to 30.1 and 47.5% at follow-up, respectively, whereas the proportion of individuals reporting poor SRH increased from 12.8 at baseline to 22.5% at follow-up.

### 3.3. SRH-Change Categories by Polypharmacy

Figure 2 and Appendix A show the baseline distribution of SRH and the SRH-change categories in participants without polypharmacy (A) and those with polypharmacy (B). Compared to participants without polypharmacy, more participants with polypharmacy were in the stable moderate category (27.5 vs. 13.5%), stable low category (6.4 vs. 2.1%), decline category (30.5 vs. 26.9%), and the improvement category (16.2 vs. 11.4%). Conversely, more participants without polypharmacy were in the stable high category (46.2 vs. 19.3%).

Table 2a demonstrates the baseline characteristics of the SRH-change categories in individuals without polypharmacy. Across all categories, the mean age (SD) ranged from 77.5 (5.5) to 79.7 (6.6). The stable high category included 349 participants (46.2%), the stable moderate category included 102 participants (13.5%), the stable low category included 16 participants (2.1%), the decline category included 203 participants (26.9%), and the improvement category included 86 participants (11.4%). The stable moderate and stable low change categories included more females (72.5% and 81.2%, respectively) compared to 52.2–55.8% in the other categories. In the stable high category, 28.1% consumed alcohol regularly, and 43.3% were physically active more than five times a week compared to 12.5–18.2% and 18.8–34.9%, respectively, in the other categories. The stable low category had the highest proportion of individuals living with at least five comorbidities (43.8%) compared to 11.7–23.5% in the other categories.

Table 2b demonstrates the baseline characteristics of SRH-change categories in individuals with polypharmacy. Participants in this group had a mean age (SD) ranging from 78.9 (5.7) to 81.0 (6.0). The stable high category included 130 participants (19.3%), the stable moderate category included 185 participants (27.5%), the stable low category included 43 participants (6.4%), the decline category included 205 participants (30.5%), and the improvement category included 109 participants (16.2%). In the improvement and stable high categories, 40% were females compared to 53.7–65.1% in the other categories. In the stable high category, 27.7% consumed alcohol regularly, and 36.2% were physically active more than five times a week compared to 15.1–19.3% and 18.6–21.1%, respectively, in the other categories. About half of the participants were living with at least five comorbidities in the stable high, stable moderate, and decline categories compared to 60.5–64.2% in the other SRH-change categories.

Differences between participants with polypharmacy and those without were more discernible across categories rather than within the individual SRH-change categories. (Table 2) Within the individual SRH-change categories, participants in the polypharmacy group were less often females except in the decline category, were less physically active, and had higher BMI values. Individuals with polypharmacy across all categories had a higher number of comorbidities (CCI ≥5; 45.4–64.2%) compared to individuals without polypharmacy (11.7–43.8%). Further analysis of participants with polypharmacy in both study visits showed an increase in the number of comorbidities as well as medications over the course of the follow-up period. Participants in the improvement category, however, had the lowest increase compared to the other categories (Appendix A).

### 3.4. Polypharmacy and SRH-Change Categories

The association between polypharmacy and SRH-change categories (Table 3) demonstrates that compared to the stable high category, polypharmacy increased the odds of being in the stable moderate category (OR 3.55; 95% CI [2.43–5.20]), stable low category (OR 3.32; 95% CI [1.65–6.70]), decline category (OR 1.87; 95% CI [1.34–2.62]), and improvement category (OR 2.01; 95% CI [1.33–3.05]) in adjusted model 1. The effect estimates were very similar when adding age as an additional covariable in adjusted model 2.

In the sensitivity analysis stratified by gender (Appendix A), the odds of being in stable categories were comparable between genders. Males showed higher odds of being in the improvement category than females (OR 2.58; 95% CI [1.42–4.67] vs. OR 1.51; 95% CI [0.82–2.80]), whereas females showed higher odds of being in the decline category than males (OR 2.64; 95% CI [1.62–4.30] vs. OR 1.30; 95% CI [0.81–2.10]).

## 4. Discussion

The polypharmacy prevalence in our study was 47.1% at baseline. We could identify five SRH-change categories in older individuals with a mean age of 79.1 (6.1) years. The majority of participants were in the stable high category (33.5%), followed by the decline category (28.6%), the stable moderate category (20.1%), the improvement category (13.7%), and lastly, the stable low category (4.1%). Almost half of the individuals without polypharmacy were in the stable high category (46.2%) compared to only about one-fifth (19.3%) of the individuals with polypharmacy. Furthermore, within the individual categories, participants with polypharmacy were less often females, were less physically active, had higher BMI values, and had a higher number of comorbidities. The adjusted multinomial regression model showed that polypharmacy was associated with increased odds of belonging to other categories than to the stable high category, with odds ratios ranging from 1.9 to 3.6. 

The polypharmacy prevalence reported in our study (47.1%) was comparable to the combined polypharmacy and hyperpolypharmacy prevalence of 48% reported in the ESTHER study conducted in Saarland, Germany, despite the younger age range of that sample (mean age 70 years) [28]. It was also comparable to the prevalence reported by Junius-Walker et al. (53.7%) in a sample of outpatients with a mean age of 77 years in the cities of Leipzig and Hannover, Germany [29].

We were able to identify five SRH-change categories; stable high, stable moderate, and stable low in addition to a decline category and an improvement category. Previous studies investigating SRH in older adults identified only three change categories (no change, improvement, and decline), with the majority of the participants showing no change in SRH over the follow-up period, which is in agreement with the results of our study when the three stable categories are combined [30,31,32]. Two of the aforementioned studies reported a higher proportion of individuals in the improvement category than in the decline category [30,32]. This could be due to the lower mean age of the participants in both studies (67.7 and 71.8 years) and longer follow-up period (nine years) [30]. On the other hand, the third study, in which the follow-up period and the mean age of the participants were comparable to ours, reported a similar distribution of individuals among SRH-change categories [31].

In our study, the highest number of participants was in the stable high category, reflecting the notion of successful aging by maintaining a high level of SRH as described by Rowe and Kahn [33]. These participants are the ones with the lowest prevalence of comorbidities and polypharmacy in our study. They are most likely the ones without functional impairment [34]. As SRH also encompasses the psychosocial aspect of health [35], the consistent rating of health as good could be an indication of positive social relationships and engagement in social activities [36]. Self-perception of aging has also been postulated to affect SRH, as individuals associating aging with positive outcomes are prompted to be more physically active with increasing age, which prevents their health from declining [37].

Conversely, the stable moderate and stable low categories were characterized by a high prevalence of comorbidities and polypharmacy at baseline in our study. Both were found to be associated with moderate-to-poor SRH [18]. The highest proportion of participants with BMI ≥30 kg/m^2^ and the lowest frequency of physical activity was found in these categories as well. Physical activity was found to affect SRH in older adults positively [38]. Therefore, the lack of physical activity in combination with multimorbidity could in part explain the stable moderate or poor SRH across both visits. This is further aggravated by the inverse relationship between SRH and BMI described in a sample of Swedish adults [39].

The decline category is a complex one reflecting the combination of senescence and pathological SRH decline with advancing age. SRH decline could be attributed to the tendency to develop more comorbidities as well as increased functional impairment [40]. Therefore, individuals rate their health progressively worse as they age [41]. 

Despite having a high number of comorbidities at baseline, individuals in the improvement category were most likely functionally independent and more physically active [30]. Possible explanations for SRH improvement could be that the individuals’ health status actually improved, and in turn, they rated their health better than before; that they perceived their health to be better compared to their peers [42]; or that they adapted to their comorbidities and learnt how to live with them [41,43].

Despite SRH improvement being favorable per se, polypharmacy was associated with elevated odds of being in the improvement category. A possible explanation could be that individuals in the improvement category had poorer SRH at baseline in comparison to participants in the stable high category. This is in line with the findings that polypharmacy is associated with worse SRH [18]. Moreover, it was previously shown that changes in pharmacodynamics and pharmacokinetics with increasing age could render older adults more prone to adverse effects in the presence of polypharmacy, which emphasizes the importance of careful drug dosing in old age [44]. It could be argued that participants in the improvement category experience less adverse effects due to careful drug selection and dosing and therefore rate their health better despite having polypharmacy. 

Studies addressing the association between polypharmacy and SRH-change categories are scarce. Compared to the existing literature, our results that polypharmacy is associated with the belonging to SRH-change categories other than the stable high category is in accordance with the results of previous studies that assessed the association between polypharmacy and HRQoL [21,45]. This association persisted after adjusting for CCI categories, implying that it is independent of multimorbidity.

Regarding gender differences, females had higher odds of being in the decline category in comparison to males. This could be explained in part by females’ tendency to consider non-life threatening chronic conditions such as hypertension as well as non-health-related life events in their assessment of health [46]. Conversely, males’ exposure to serious and potentially life-threatening medical conditions tends to play a larger role in reporting their SRH [46]. It could be argued that males have higher odds of reporting better SRH after they have passed the acute episode of a serious illness. Furthermore, females were found to experience more drug–drug interactions compared to males due to higher medication intake as well as sex-related differences in pharmacokinetics in old age [47,48].

There are several pathways through which polypharmacy might negatively influence the progression of SRH. The mere consumption of several medications and experiencing their side effects could lead older adults to perceive their health as poor [49]. Moreover, polypharmacy was found to be associated with functional impairment in older adults reflected in difficulties performing basic and/or instrumental activities of daily life, which in turn was found to be an indicator of worsening SRH [50,51]. This is further aggravated by the association of polypharmacy with geriatric syndromes including urinary incontinence, falls, cognitive impairment, and frailty, causing them to be more dependent in performing their daily activities [52].

Non-adherence could also contribute to the negative effect of polypharmacy on SRH. Non-adherence refers to deviations from the prescribed treatment, including overuse, underuse, or incorrect use of medications. It could be attributed to regimen complexity, forgetfulness, or side effects of the medications [53]. It was reported that individuals with polypharmacy have lower adherence to prescribed medications [54]. This increases the likelihood of complications which reduces HRQoL and could prompt physicians to prescribe more medications, leading to a prescribing cascade [19,55]. It could also be discussed that the resulting poor SRH may be due to lack of control of symptoms resulting from non-adherence.

Our study benefited from multiple strengths. First, this study included a large cohort of 1428 community-dwelling older adults and longitudinal data with a follow-up period of four years. Second, we included all regularly taken medications including OTC and did not restrict them only to prescription medications. Third, the data utilized for this study were extracted from primary BIS data complemented by secondary claims data. This study was not without limitations. First, as the information about medication was self-reported, it was prone to recall bias. However, we tried to minimize bias through asking participants to bring their medication lists and packages to thoroughly assess their medication intake. Second, the polypharmacy definition we used was based on a numerical threshold and did not address medication appropriateness. However, this enabled us to compare our results to other studies, as the definition we used is the most commonly used definition of polypharmacy. Third, we did not include cognitive ability as an inclusion criterion because cognitive impairment was assessed only during the SRH follow-up visit. However, there were no major differences between those with and without cognitive impairment in the distribution among the SRH-change categories [56]. Fourth, we were not able to consider the individual severity of comorbidities as a confounder although the CCI incorporates clinical weights to control for the severity of the conditions. This might have affected the association between polypharmacy and SRH-change categories, as varying disease severity could have affected the SRH of participants with the same disease differently. Lastly, excluded participants had a higher prevalence of polypharmacy and reported their health less often as good, which might have led to an underestimation of the association.

## 5. Conclusions

Although polypharmacy is recognized as a public health challenge in older individuals, its effect on SRH-change independent of comorbidities is widely unrecognized. Our finding should prompt physicians to implement strategies to alleviate polypharmacy, such as medication reconciliation and possible deprescription as well as utilization of non-pharmacological approaches to control factors contributing to the development of multimorbidity (e.g., BMI). Simplification of drug regimens and the development of single-pill combinations for the most commonly occurring comorbidities in older adults could be an additional strategy to increase adherence and subsequently decrease potential drug–drug interactions and improve SRH. To empower this vulnerable group, the aforementioned strategies should be implemented while keeping individual preferences of older adults at the forefront of the decision-making process. This could then lead not only to positive changes to their physical but also to their psychosocial wellbeing.

Additional factors other than the number of medications could affect the association between polypharmacy and the SRH-change categories, such as the type and appropriateness of medications. In this respect, future research should address the improvement and decline categories in greater detail while taking gender differences into consideration. 

## Figures and Tables

**Figure 1 ijerph-20-04159-f001:**
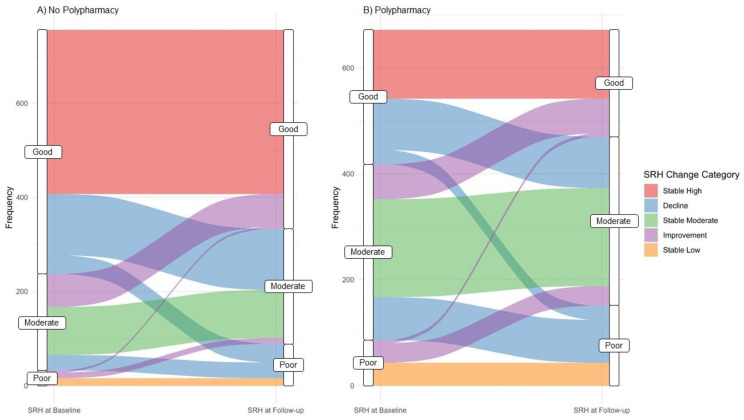
Transition in SRH reporting at baseline and follow-up in participants (**A**) without polypharmacy and (**B**) with polypharmacy. The flow of participants over the observation period represents the SRH-change categories. The width of the lines is proportional to the number of participants.

**Figure 2 ijerph-20-04159-f002:**
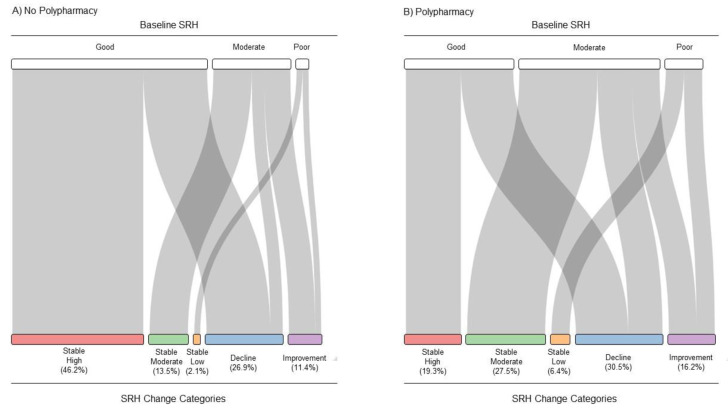
Distribution of self-rated health (SRH) at baseline (top part) of participants (**A**) without polypharmacy and (**B**) with polypharmacy and their assignment to SRH-change categories (bottom part) over the period of four years. The colors correspond to SRH-change categories in Figure 1.

**Table 1 ijerph-20-04159-t001:** Main characteristics of the study population by polypharmacy status.

Variable	Category	No Polypharmacy	Polypharmacy	Total (N = 1428)
		756 (52.9%)	672 (47.1%)	
Sociodemographic Factors				
Age Mean (SD)		78.3 (6.1)	79.9 (5.9)	79.1 (6.1)
Gender N (%)	Female	426 (56.3%)	345 (51.3%)	771 (54.0%)
SRH Level at baseline N (%)	Good	519 (68.7%)	254 (37.8%)	773 (54.1%)
	Moderate	205 (27.1%)	332 (49.4%)	537 (37.6%)
	Poor	32 (4.2%)	86 (12.8%)	118 (8.3%)
Income (in EUR) N (%)	<1000	215 (28.4%)	194 (28.9%)	409 (28.6%)
	1000–1999	388 (51.3%)	366 (54.5%)	754 (52.8%)
	≥2000	49 (6.5%)	31 (4.6%)	80 (5.6%)
	Missing	104 (13.8%)	81 (12.1%)	185 (13%)
CASMIN N (%)	Low	435 (57.5%)	412 (61.3%)	847 (59.3%)
	Intermediate	155 (20.5%)	133 (19.8%)	288 (20.2%)
	High	162 (21.4%)	124 (18.5%)	286 (20%)
	Missing	4 (0.5%)	3 (0.4%)	7 (0.5%)
Having a Partner N (%)		457 (60.4%)	403 (60.0%)	860 (60.2%)
Lifestyle Factors				
Frequency of Alcohol Consumption N (%)	Less than once a month	290 (38.4%)	317 (47.2%)	607 (42.5%)
	≤2 times per week	300 (39.7%)	221 (32.9%)	521 (36.5%)
	Regularly	166 (22.0%)	130 (19.3%)	296 (20.7%)
	Missing	0	4 (0.6%)	4 (0.3%)
Physical Activity N (%)	Less than once a week	104 (13.8%)	208 (31%)	312 (21.8%)
	1–5 times per week	376 (49.7%)	305 (45.4%)	681 (47.7%)
	More than 5 times per week	275 (36.4%)	158 (23.5%)	433 (30.3%)
	Missing	1 (0.1%)	1 (0.1%)	2 (0.1%)
BMI (in kg/m^2^) N (%)	<25	228 (30.2%)	127 (18.9%)	355 (24.9%)
	25–<30	367 (48.5%)	321 (47.8%)	688 (48.2%)
	≥30	161 (21.3%)	223 (33.2%)	384 (26.9%)
	Missing	0	1 (0.1%)	1 (0.1%)
Medical Status				
CCI Category N (%)	0	119 (15.7%)	19 (2.8%)	138 (9.7%)
	1–2	295 (39%)	115 (17.1%)	410 (28.7%)
	3–4	196 (25.9%)	176 (26.2%)	372 (26.1%)
	≥5	136 (18.0%)	360 (53.6%)	496 (34.7%)
	Missing	2 (0.3%)	10 (1.3%)	12 (0.8%)

SD, standard deviation; SRH, self-rated health; CASMIN, Comparative Analysis of Social Mobility in Industrial Nations; BMI, body mass index in kg/m^2^; CCI, Charlson Comorbidity Index.

**Table 2 ijerph-20-04159-t002:** (**a**) Baseline characteristics of individuals without polypharmacy by SRH-change category. (**b**) Baseline characteristics of individuals with polypharmacy by SRH-change category.

(a) No Polypharmacy Group							
Variable	Category	SRH-Change Category					Total (N = 756)
		Stable High	Stable Moderate	Stable Low	Decline	Improvement	
		349 (46.2%)	102 (13.5%)	16 (2.1%)	203 (26.9%)	86 (11.4%)	
Sociodemographic Factors							
Age Mean (SD)		77.5 (5.5)	78.2 (6.4)	79.0 (6.5)	79.7 (6.6)	78.4 (6.1)	78.3 (6.1)
Gender N (%)	Female	185 (53.0%)	74 (72.5%)	13 (81.2%)	106 (52.2%)	48 (55.8%)	426 (56.3%)
Income (in EUR) N (%)	<1000	86 (24.6%)	34 (33.3%)	8 (50.0%)	60 (29.6%)	27 (31.4%)	215 (28.4%)
	1000–1999	185 (53.0%)	43 (42.2%)	6 (37.5%)	110 (54.2%)	44 (51.2%)	388 (51.3%)
	≥2000	26 (7.4%)	8 (7.8%)	1 (6.2%)	10 (4.9%)	4 (4.7%)	49 (6.5%)
	Missing	52 (14.9%)	17 (16.7%)	1 (6.2%)	23 (11.3%)	11 (12.8%)	104 (13.8%)
CASMIN N (%)	Low	198 (56.7%)	64 (62.7%)	10 (62.5%)	111 (54.7%)	52 (60.5%)	435 (57.5%)
	Intermediate	73 (20.9%)	17 (16.7%)	5 (31.2%)	47 (23.2%)	5 (31.2%)	155 (20.5%)
	High	78 (22.3%)	21 (20.6%)	1 (6.2%)	42 (20.7%)	20 (23.3%)	162 (21.4%)
	Missing	0	0	0	3 (1.5%)	1 (1.2%)	4 (0.5%)
Having a Partner N (%)		229 (65.6%)	55 (53.9%)	8 (50.0%)	112 (55.2%)	53 (61.6%)	457 (60.4%)
Lifestyle Factors							
Frequency of Alcohol Consumption N (%)	Less the once a month	108 (30.9%)	48 (47.1%)	12 (75.0%)	84 (41.4%)	38 (44.2%)	290 (38.4%)
	≤2 times per week	143 (41.0%)	40 (39.2%)	2 (12.5%)	82 (40.4%)	33 (38.4%)	300 (39.7%)
	Regularly	98 (28.1%)	14 (13.7%)	2 (12.5%)	37 (18.2%)	15 (17.4%)	166 (22.0%)
	Missing	0	0	0	0	0	0
Physical Activity N (%)	Less than once a week	35 (10.0%)	16 (15.7%)	5 (31.2%)	29 (14.3%)	19 (22.1%)	104 (13.8%)
	1–5 times per week	162 (46.4%)	56 (54.9%)	8 (50.0%)	113 (55.7%)	37 (43.0%)	376 (49.7%)
	More than 5 times per week	151 (43.3%)	30 (29.4%)	3 (18.8%)	61 (30.0%)	30 (34.9%)	275 (36.4%)
	Missing	1 (0.3%)	0	0	0	0	1 (0.1%)
BMI (in kg/m^2^) N (%)	<25	107 (30.7%)	36 (35.3%)	4 (25.0%)	57 (28.1%)	24 (27.9%)	228 (30.2%)
	25–<30	175 (50.1%)	45 (44.1%)	8 (50.0%)	97 (47.8%)	42 (48.8%)	367 (48.5%)
	≥30	67 (19.2%)	21 (20.6%)	4 (25.0%)	49 (24.1%)	20 (23.3%)	161 (21.3%)
	Missing	0	0	0	0	0	0
Medical Status							
CCI Category N (%)	0	74 (21.2%)	12 (11.8%)	0	25 (12.3%)	8 (9.3%)	119 (15.7%)
	1–2	143 (41.0%)	36 (35.3%)	7 (43.8%)	79 (38.9%)	30 (34.9%)	295 (39.0%)
	3–4	85 (24.4%)	30 (29.4%)	2 (12.5%)	51 (25.1%)	28 (32.6%)	196 (25.9%)
	≥5	41 (11.7%)	24 (23.5%)	7 (43.8%)	44 (21.7%)	20 (23.3%)	136 (18.0%)
	Missing	6 (1.7%)	0	0	4 (2.0%)	0	10 (1.3%)
**(b) Polypharmacy Group**							
**Variable**	**Category**	**SRH-Change Category**					**Total** **(N = 672)**
		**Stable High**	**Stable Moderate**	**Stable Low**	**Decline**	**Improvement**	
		**130 (19.3%)**	**185 (27.5%)**	**43 (6.4%)**	**205 (30.5%)**	**109 (16.2%)**	
Sociodemographic Factors							
Age Mean (SD)		79.7 (5.9)	79.4 (5.9)	79.8 (6.0)	81.0 (6.0)	78.9 (5.7)	79.9 (6.0)
Gender N (%)	Female	52 (40.0%)	111 (60.0%)	28 (65.1%)	110 (53.7%)	44 (40.4%)	345 (51.3%)
Income (in EUR) N (%)	<1000	33 (25.4%)	64 (34.6%)	16 (37.2%)	49 (23.9%)	32 (29.4%)	194 (28.9%)
	1000–1999	68 (52.3%)	95 (51.4%)	21 (48.8%)	123 (60.0%)	59 (54.1%)	366 (54.5%)
	>=2000	9 (6.9%)	8 (4.3%)	2 (4.7%)	8 (3.9%)	4 (3.7%)	31 (4.6%)
	Missing	20 (15.4%)	18 (9.7%)	4 (9.3%)	25 (12.2%)	14 (9.7%)	81 (12.1%)
CASMIN N (%)	Low	77 (59.2%)	117 (63.2%)	29 (67.4%)	123 (60.0%)	66 (60.6%)	412 (61.3%)
	Intermediate	22 (16.9%)	36 (19.5%)	9 (20.9%)	42 (20.5%)	24 (22.0%)	133 (19.8%)
	High	31 (23.8%)	31 (16.8%)	5 (11.6%)	39 (19.0%)	18 (16.5%)	124 (18.5%)
	Missing	0	1 (0.5%)	0	1 (0.5%)	1 (0.9%)	3 (0.4%)
Having a Partner N (%)		82 (63.1%)	107 (57.8%)	22 (51.2%)	121 (59.0%)	71 (65.1%)	403 (60.0%)
Lifestyle Factors							
Frequency of Alcohol Consumption N (%)	Less the once a month	44 (33.8%)	107 (57.8%)	27 (62.8%)	83 (40.5%)	56 (51.4%)	317 (47.2%)
	≤2 times per week	49 (37.7%)	49 (26.5%)	8 (18.6%)	84 (41.0%)	31 (28.4%)	221 (32.9%)
	Regularly	36 (27.7%)	28 (15.1%)	8 (18.6%)	37 (18.0%)	21 (19.3%)	130 (19.3%)
	Missing	1 (0.8%)	1 (0.5%)	0	1 (0.5%)	1 (0.9%)	4 (0.6%)
Physical Activity N (%)	Less than once a week	27 (20.8%)	58 (31.4%)	24 (55.8%)	65 (31.7%)	34 (31.2%)	208 (31.0%)
	1–5 times per week	56 (43.1%)	87 (47.0%)	11 (25.6%)	98 (47.8%)	53 (48.6%)	305 (45.4%)
	More than 5 times per week	47 (36.2%)	39 (21.1%)	8 (18.6%)	42 (20.5%)	22 (20.2%)	158 (23.5%)
	Missing	0	1 (0.5%)	0	0	0	1 (0.1%)
BMI (kg/m^2^) N (%)	<25	30 (23.1%)	32 (17.3%)	7 (16.3%)	39 (19.0%)	19 (17.4%)	127 (18.9%)
	25–<30	68 (52.3%)	85 (45.9%)	19 (44.2%)	94 (45.9%)	55 (50.5%)	321 (47.8%)
	≥30	32 (24.6%)	68 (36.8%)	17 (39.5%)	71 (34.6%)	35 (32.1%)	223 (33.2%)
	Missing	0	0	0	1 (0.5%)	0	1 (0.1%)
Medical Status							
CCI Category N (%)	0	5 (3.8%)	7 (3.8%)	0	6 (2.9%)	1 (0.9%)	19 (2.8%)
	1–2	36 (27.7%)	24 (13.0%)	8 (18.6%)	34 (16.6%)	13 (11.9%)	115 (17.1%)
	3–4	30 (23.1%)	58 (31.4%)	9 (20.9%)	54 (26.3%)	25 (22.9%)	176 (26.2%)
	≥5	59 (45.4%)	95 (51.4%)	26 (60.5%)	110 (53.7%)	70 (64.2%)	360 (53.6%)
	Missing	0	1 (0.5%)	0	1 (0.5%)	0	2 (0.3%)

SD, standard deviation. SRH, self-rated health; CASMIN, Comparative Analysis of Social Mobility in Industrial Nations; BMI, body mass index in kg/m^2^; CCI, Charlson Comorbidity Index.

**Table 3 ijerph-20-04159-t003:** Multinomial regression model showing the association between polypharmacy and SRH-change categories.

	SRH-Change Categories					
	Stable High	Stable Moderate	Stable Low	Decline	Improvement	
N (%)						
Polypharmacy						<0.001
Yes	130 (19.3)	185 (27.5)	43 (6.4)	205 (30.5)	109 (16.2)	
No	349 (46.2)	102 (13.5)	16 (2.1)	203 (26.9)	86 (11.4)	
Crude Model OR (95% CI)						
Polypharmacy (Yes)	Reference	4.87 (3.56–6.67)	7.22 (3.93–13.26)	2.71 (2.05–3.59)	3.40 (2.40–4.81)	
Adjusted Model 1 ^a^ OR (95% CI)						
Polypharmacy (Yes)	Reference	3.55 (2.43–5.20)	3.32 (1.65–6.70)	1.87 (1.34–2.62)	2.01 (1.33–3.05)	
Adjusted Model 2 ^b^ OR (95% CI)						
Polypharmacy (Yes)	Reference	3.54 (2.41–5.18)	3.34 (1.65–6.76)	1.83 (1.31–2.57)	2.01 (1.33–3.06)	

^a^ Adjusted for gender, income, Comparative Analysis of Social Mobility in Industrial Nations (CASMIN), partner status, frequency of alcohol consumption, physical activity, body mass index (BMI), and Charlson Comorbidity Index (CCI). ^b^ Adjusted model 1 + age.

## Data Availability

The data presented in this study are available upon reasonable request from the corresponding author.

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
