# Peer review of "Polypharmacy and the Change of Self-Rated Health in Community-Dwelling Older Adults"

_ijerph, 2023, doi:10.3390/ijerph20054159_

Round 1
Reviewer 1 Report
The study 1) identified different “trajectories” of self-rated health (SRH) over a period of four years and 2) assesses the associations between polypharmacy and different SRH “trajectories” based on data from the Berlin Initiative Study (BIS). The authors identified five SRH “trajectories”: stable high, stable moderate, stable low, decline, and improvement. Polypharmacy was associated with being in “trajectories” other than the stable high “trajectory”. Overall, the topic of the manuscript fits well with the interests of the IJERPH. However, I have a number of questions, suggestions, and comments and my advice would be to have the authors address these in a revised version of their paper.
General
- I am not sure, if “trajectory” is the best wording for a simple difference between two measurement occasions four years apart. Of course, there can be a linear trajectory over the four years, but the authors do not (can not?) test for the form of the trajectory (linear, nonlinear). My suggestion would be, to adjust the wording into “SRH change (groups)”, for example (or maybe SRH trend?), and use this wording throughout the whole manuscript.
- The authors state in the measurement section that they collapsed the extreme SRH categories due to low numbers in the extreme categories and use SRH as a three category variable: (very) good, moderate, (very) bad). I would suggest to use this three category variable from then on throughout the whole manuscript. Instead, the authors present results for all five categories in the tables in the result section. I would suggest to present the five categories only in suppl. material for the interested reader and to present the three category variable in the manuscript (in tables, figure, text!).
Methods
- Study population:
o Why were dialysis patients and kidney transplant patients excluded?
o I’m not familiar with the BIS data, but it seems that there are more measurement occasions available – why do the authors use only two and why these two specific occasions and not the others?
- Measures:
o After SRH is introduced here in the measures section, SRH should be reported as a three category variable throughout the manuscript (see above)
o Age is reported in the measurement section (covariables) and in tables in the descriptive result section but is not mentioned in the statistical analyses section (“…adjusted for…”, page 2, lines 142 ff.) and is also not mentioned in table 3. Could the authors please clarify: does this mean that the regression models are not adjusted for age – and if so, why?
o Regarding various comorbidities in the covariable/results section: why are additional comorbidities considered to CCI and why these specific comorbidities? Would the CCI itself not be sufficient? I would suggest that the authors should describe in more detail, why they consider these additional comorbidities (or only use the CCI). Also, the other comorbidities are missing in Table 3
- Statistical analyses:
o As mentioned above, not all earlier introduced covariables are part of the multinomial regression model; why (not)?
Results
- It seems that differences between the polypharmacy group and the group without polypharmacy are not tested for statistical significance (Table 1). I would recommend to add such analyses
- Regarding Table 1: SRH should be reported with only three categories. Additionally, if authors decide to report less covariates, Table 1 should be adjusted accordingly
- Fig. 1 would be much more easier to read, if only the three categories of baseline SRH were used as it is described in the measures section; furthermore, an additional heading regarding the polypharmacy group would be helpful (A: Without Polypharmacy; B: With Polypharmacy; or something like that)
Discussion
- It would be very helpful, if authors could restructure the first part of the discussion according to the research questions/results section (meaning: 1) SRH groups, 2) Polypharmacy, 3) Association between SRH and Polypharmacy; and after that the sections about possible pathways and strengths/limitations)
Minor issues:
1. Just a suggestion: an additional schematic figure for the different SRH changes would help to get a better understanding of the different SRH groups
2. Table 3 à CASMIN in the footnote needs further explanation as it is done with BMI
3. There is something “wrong” in the second line of the conclusion section (“This finding should prompt” – something is odd with should in my version of the manuscript)
Reviewer 2 Report
This study aimed to 1) identify trajectories of self-rated health (SRH) over the period of four years and describe how they differ between older individuals with and without 53 polypharmacy and 2) assess the association between polypharmacy and different SRH 54 trajectories. The topic is interesting, the study design is well-written and the results are presented appropriately.
I have the following comments and suggestions:
Introduction:
1. Many citations used are very outdated, try to add new citations preferably from the last 5 years.
2. Does self-rated health (SRH) is same as the patient-reported outcome (PRO)? I suggest using PRO instead.
Method:
Well-written
Results:
3. Table 1. Main characteristics of the study population by polypharmacy status. Please show the p-value, so that we know if there are significant differences or not.
4. Table 2. Baseline characteristics of SRH trajectories for the study population. Are these combined results of poly and non-poly pharmacy? If yes, then what is the point of combining them?
5. Figure 1. Distribution of self-rated health (SRH) at baseline (top part) of participants A) without 210 polypharmacy and B) with polypharmacy and their assignment to SRH trajectories (bottom part) 211 over the period of four years.
I suggest writing on the top of the figure poly/non-poly pharmacy before "baseline SRH".
6. Table 3. Multinomial regression model showing the association between polypharmacy and SRH 233 trajectories.
This should be a comparison between poly and non-poly pharmacy groups in relation to SRH. Also need to display the p-value.
Conclusion:
It is a very weak conclusion. Physicians prescribe medication with a reason, from my understanding, one of the methods can be to find out what are the most common comorbidities and make pills by combining drugs for such most common comorbidities. Also, I suggest recommending patient empowerment and engagement to deal with the poly-pharmacy issues.
Reviewer 3 Report
This is a well-conducted study named “Polypharmacy and the trajectories of self-rated health in community-dwelling older adults”. In this study, the author highlighted that reducing polypharmacy could be an impactful strategy to foster favourable self-rated health progression in old age. This finding will be of interest to the readers of this journal. However, to strengthen the manuscript, it would be good to consider the following suggestions.
1. I acknowledged that the authors have discussed the reason why this study is needed in the introduction part. But, in my view, it is not obvious why SRH is important compared to other self-reported measures such as quality of life and health-related quality of life tools.
2. The authors used the “trajectories” term although it is the change in SRH between two-time points. In my view, it is better to say “Change in SRH” rather than trajectories.
It is better to name “trajectories” when the authors applied the modelling statistics using the data of > 2-time points like references 26-28.
So, it would be preferable to appropriately revise the entire manuscript (change the term if the authors agree with this suggestion-2 and then find the appropriate references rather than trajectories papers and then discuss about it in the discussion).
3. As gender plays important role in both SRH and polypharmacy, it would be good to see the results of the main analysis according to gender (as a sensitivity analysis in the supplementary materials).
4. As per supplementary figure S1, I noticed that there was some missing and some lost to follow-up. So, it would be good to present the differences in the baseline characteristics between 1428 participants (the included participants of this study) and excluded participants in the supplementary materials and discuss about it if you find some significant results.
5. In my view, the discussion needs to report the direct clinical and public health implications and recommendations (about one paragraph) based on the authors’ novel findings of this study.
Round 2
Reviewer 1 Report
The present manuscript is a revision of a manuscript that presented a study that 1) identified different self-rated health (SRH) change groups over a period of four years and 2) assessed the associations between polypharmacy and different SRH change groups based on data from the Berlin Initiative Study (BIS). The authors have responded to my earlier comments. Some changes have been made, many parts of the manuscript are now clearer and the authors included some of the missing details I was seeking. Although the paper is much improved, I have a few remaining thoughts about this manuscript, which I hope the authors will take the time to address:
First, regarding Figure 1: in my opinion, the “increase” in the stable moderate group suggests an improvement – my suggestion would be to use the same outlet as in Figure 2 (from top to bottom and not left to right); also, I would like to recommend to use different colors which are still possible to see and especially to keep apart in a printout of the manuscript; please avoid the wording “trajectories” in the figure description; maybe change the order of the Figure to Polypharmacy first and then on the right side no polypharmacy (as that is the order in Table 1 and 2 – or change the order in Table 1 and 2; however I would recommend to decide for one order and use this order throughout the whole manuscript (text, figures, tables)).
Second, regarding Figure 2: I would like to suggest, to use the same colors in Figure 2 as introduced in Figure 1 – this change would make it much easier to understand for the reader, in my opinion
Finally, regarding Table 2 à I would like to suggest to make it clearer, that the top part of the Table is for no polypharmacy and the second part for polypharmacy (It is already stated in the heading of the Table but I would like to suggest to also incorporate this information into the table itself); also, there is a heading within the first half of the Table (“Polypharmacy and SRH”) which can be omitted, in my opinion
Very minor issues:
- There is a “%” too much on p. 5, line 187 (…from 68.7 at baseline to 56.0%...)
- In Table 3, there is a “<0.001” in the line of “polypharmacy”?
- In the first paragraph of the discussion (summary), I would recommend to switch the information regarding polypharmacy and SRH categories (start with polypharmacy, then SRH (and after that the association) to match the structure of the following discussion)
Reviewer 2 Report
I am satisfied with the response and the rebuttal submitted.
Reviewer 3 Report
Well done. Authors. Congratulations.
